# Run-Up Simulation of a Semi-Floating Ring Supported Turbocharger Rotor Considering Thrust Bearing and Mass-Conserving Cavitation

Christian Ziese *, Cornelius Irmscher, Steffen Nitzschke, Christian Daniel and Elmar Woschke

Institute of Mechanics, Otto von Guericke University Magdeburg, 39106 Magdeburg, Germany; cornelius.irmscher@ovgu.de (C.I.); steffen.nitzschke@ovgu.de (S.N.); christian.daniel@ovgu.de (C.D.); elmar.woschke@ovgu.de (E.W.)
* Correspondence: christian.ziese@ovgu.de; Tel.: +49-391-67-52885

**Abstract:** The vibration behaviour of turbocharger rotors is influenced by the acting loads as well as by the type and arrangement of the hydrodynamic bearings and their operating condition. Due to the highly non-linear bearing behaviour, lubricant film-induced excitations can occur, which lead to sub-synchronous rotor vibrations. A significant impact on the oscillation behaviour is attributed to the pressure distribution in the hydrodynamic bearings, which is influenced by the thermo-hydrodynamic conditions and the occurrence of outgassing processes. This contribution investigates the vibration behaviour of a floating ring supported turbocharger rotor. For detailed modelling of the bearings, the Reynolds equation with mass-conserving cavitation, the three-dimensional energy equation and the heat conduction equation are solved. To examine the impact of outgassing processes and thrust bearing on the occurrence of sub-synchronous rotor vibrations separately, a variation of the bearing model is made. This includes run-up simulations considering or neglecting thrust bearings and two-phase flow in the lubrication gap. It is shown that, for a reliable prediction of sub-synchronous vibrations, both the modelling of outgassing processes in hydrodynamic bearings and the consideration of thrust bearing are necessary.

**Keywords:** run-up simulation; semi-floating ring bearing; thrust bearing; two-phase flow cavitation; oil-whirl and oil-whip

## 1. Introduction

Turbochargers have become indispensable in combustion engines due to economic, ecological and efficiency reasons. The engine's charge exchange work is no longer sufficient to satisfy the required demands, so additional charging is necessary. The rotor's principle design consists of a compressor and turbine wheel taken up by a common shaft. Due to the flow of exhaust gases to the turbine wheel, the kinetic energy of the gases is used to set the rotor in a rotational motion. Simultaneously on the compressor side, the fresh air is compressed and supplied to the combustion chamber. Consequently, the main aim of turbocharging is to increase the power and combustion efficiency of the engine by increasing the mean pressure while reducing fuel consumption if possible.

An essential factor for compliance with the requirements is the support of the rotor. This is preferably achieved by using hydrodynamic bearings. Compared to roller bearings, hydrodynamic bearings have a simpler and more cost-efficient design as well as more favourable thermo-hydrodynamic operating conditions. A disadvantage is their highly non-linear behaviour, which can even lead to additional rotor excitations in terms of sub-synchronous vibration [1–11]. Woschke et al. [1,2] examined the excitation mechanism in context with the current natural frequencies and damping of the rotor. By solving the eigenvalue problem at the current operating point, including the non-linear bearing properties, a Campbell diagram with the corresponding damping is plotted and the rotor

excitation via the half-whirl frequency is evaluated over the entire run-up. It is shown that an oil-whip can occur if a weakly damped natural frequency is excited by an oil-whirl. The current rotor natural frequencies depend on the mass and stiffness distributions of the rotor as well as on the stiffness and damping properties of the individual bearings [2]. With the occurrence of oil-whips, increased vibration amplitudes, wear and even failure of the rotor can occur. For this reason, a reliable prediction of the critical sub-synchronous vibrations is necessary.

Concerning the stiffness and damping properties of hydrodynamic bearings, the kinematics of shaft and housing, the temperature distribution and outgassing processes have significant influences. Temperature changes lead to thermal gap changes. In addition, the lubricant properties such as the viscosity are non-linearly dependent on it. Due to thermal boundary conditions and the weak thermal conductivity of common lubricating oils, the viscosity can vary in all three spatial directions, which must be taken into account owing to the aforementioned non-linear temperature dependence. For this reason, Dowson derived the generalised Reynolds equation [12], which allows the consideration of a viscosity that varies over the gap height. In the following years, thermo-hydrodynamic phenomena were thoroughly investigated, which is comprehensively documented in the reviews [13–16]. In more recent publications, a transient consideration of the three-dimensional thermal phenomena is increasingly carried out (see, e.g., [17–19]). This is a requirement for the investigation of vibration phenomena.

Outgassing processes describe the presence of gas in the lubricating film with the consequence of developing a multi-phase flow [20]. In hydrodynamic bearings, vapour and gas cavitation as well as aeration can occur in particular. Vapour cavitation appears if the hydrodynamic pressure falls below the vapour pressure so that the lubricant changes its aggregate state. In contrast, gas cavitation is a diffusion-induced process in which the gas dissolved in the lubricant undergoes a phase transition if the pressure falls below the saturation pressure. Besides vapour and gas cavitation, air from the environment can also penetrate the lubrication gap via the bearing edges (aeration) [21]. With the occurrence of two-phase flow, a partially filled lubricating gap is established, which leads to a softer bearing behaviour and consequently favours the appearance of sub-synchronous rotor vibrations. For modelling outgassing processes, the two-phase model is applied in this contribution. It is a mass-conserving cavitation model, which makes it possible to consider outgassing processes depending on the hydrodynamic pressure and the lubricant film temperature [3,22–25]. For completeness, the cavitation theories according to Jakobsson Floberg and Olsson (JFO-model) or its efficient numerical implementation by Elrod [11,26–29] and the model of bubble dynamics [30] may also be mentioned.

In addition to the stiffness and damping behaviour of the individual bearings, the arrangement and type of bearings also influence the current rotor natural frequency. The radial displacement of the rotor is mainly supported by journal bearings. Consequently, this bearing type contributes to the stiffness and damping in horizontal and vertical rotor direction, while the tilting stiffness is limited. In terms of tilting stiffness and damping, thrust bearings contribute essentially. The rotor tilting results in an asymmetrical pressure distribution at the thrust bearing so that a torque is created, which counteracts the rotor tilting. Thus, the thrust bearing provides a further part to the stiffness and damping, which can influence the current natural frequencies of the rotor. Investigations of the influence of thrust bearings on a full-floating ring supported turbocharger rotor are carried out in [31–36]. A variation of the axial load and thrust bearing design parameters is made in [32]. The simulations show that with increasing axial load, the start frequency of the oil-whip can be shifted to higher speed ranges. This is because smaller lubrication gaps can occur at the thrust bearing so that the hydrodynamic pressure and consequently the stiffness and damping of the bearing increase. In comparison, the number of bearing segments seems to have only a minor influence.

This contribution deals with the influence of outgassing processes as well as the impact of the thrust bearing on the occurrence of sub-synchronous vibrations for a semi-

floating ring supported turbocharger rotor. Due to the design, the rotor's centre of gravity is close to the turbine bearing, which results in a general tendency to high rotor tilting during operation. Under these circumstances, it is essential to consider the thrust bearing in order to predict the sub-synchronous vibrations and the rotor amplitudes reliably. To investigate the individual effects on the rotor's response behaviour, various models of the hydrodynamic bearings are chosen and their influences are compared with the results of the run-up measurement. The level of detail includes the Half-Sommerfeld solution without thrust bearing up to the two-phase model with thrust bearing. For detailed modelling of the thermo-hydrodynamic states of the journal bearings, the Reynolds equation with mass-conserving cavitation according to the two-phase model and the three-dimensional energy equation for the temperature distribution in the lubricating film as well as the heat conduction equation for the shaft, bushing and housing are solved. Regarding the thrust bearing, the Reynolds equation with centrifugal flow and mass-conservation cavitation is solved.

## 2. Theoretical Fundamentals

The examined rotor consists of an elastic shaft, the compressor and turbine wheel, as well as the sealing disk, auxiliary bearing and thrust ring (cf. Figure 1). To model the elastic deformations of the shaft, the Finite-Element-Method (FEM) according to the Timoshenko beam theory [37] is applied. The FE-nodes are positioned at relevant shaft shoulders as well as at the bearings in order to determine the lubrication gap. Furthermore, the components assembled on the shaft can be defined as rigid bodies and their inertia properties are assigned to the corresponding FE-nodes. The total mass of the rotor is $m_{Rot}$ = 20 kg and its centre of gravity is located close to the turbine bearing. To take the unbalance condition of the rotor into account, the compressor and turbine wheel have an unbalance distribution $(u_C, u_T)$ with an angular offset. In addition to the rotor, the compressor and turbine side floating rings exist as separate rigid bodies within a multi-body environment. The interaction between the individual components is realised via the resulting forces and torques of the hydrodynamic films. These are marked blue in the figure. As a result, there is an interaction between housing and bushing (outer lubrication gap) and between bushing and shaft (inner lubrication gap) at the journal bearing. The forces and torques occurring at the thrust bearings act between the bearing housing (inertial system) and the thrust ring or auxiliary bearing.

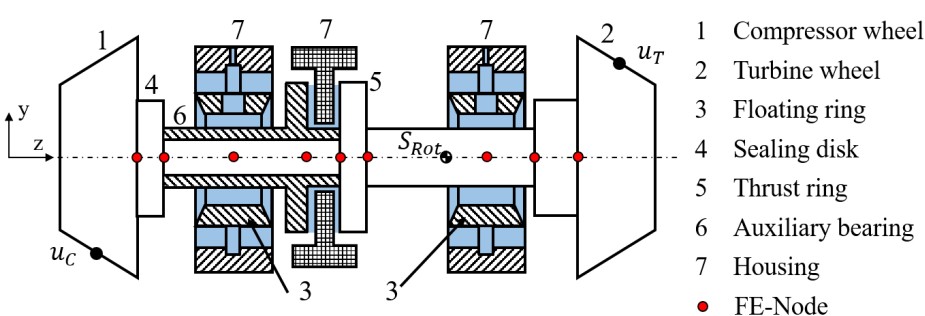

**Figure 1.** Schematic representation of the rotor model.

For the radial support of the rotor, floating ring bearings with outer squeeze-film damping are used. Due to the prevention of the rotational movement of the bushing around the longitudinal rotor axis, a hydrodynamic pressure can only be generated at the outer gap if there is a squeezing of the lubricant. To ensure a sufficient lubricant supply, the outer film has a circumferential groove, so the oil is distributed evenly over the entire circumference. Furthermore, the oil can flow through the connecting channels of the bushing to the inner lubrication gap and also build up a hydrodynamic pressure. Accordingly, the lubricating films are coupled to each other, whereby the lubricant can

flow either to the inner or to the outer gap, depending on the operating bearing condition. It should be mentioned that the inner lubrication gap has a multi-lobe geometry consisting of three segments and both the inner and outer gaps are unsealed. A schematic illustration of the bearing with considered fluid flows is shown in Figure 2.

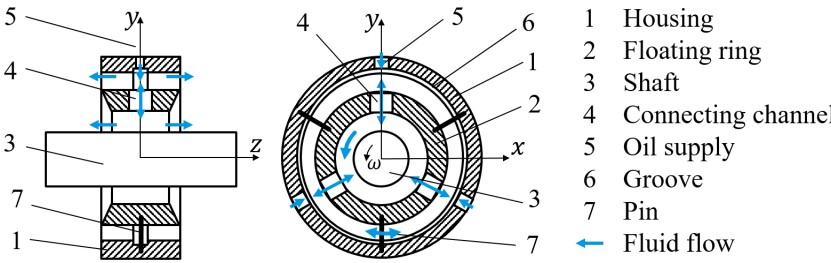

**Figure 2.** Schematic representation of the floating ring bearing with considered fluid flows.

For the axial support of the rotor, thrust bearings are used. With regard to the design, each segment has a wedge, flat and groove area, whereby the oil is supplied through holes in each groove area. To avoid the oil flowing off at the outer edge, the thrust bearings are additionally sealed. A sealing effect is achieved by the fact that the outer bearing edge has no contour. Due to the turbocharger design, the axial lubrication films on the compressor and turbine sides can be considered separately. A schematic design of the thrust bearing can be seen in Figure 3.

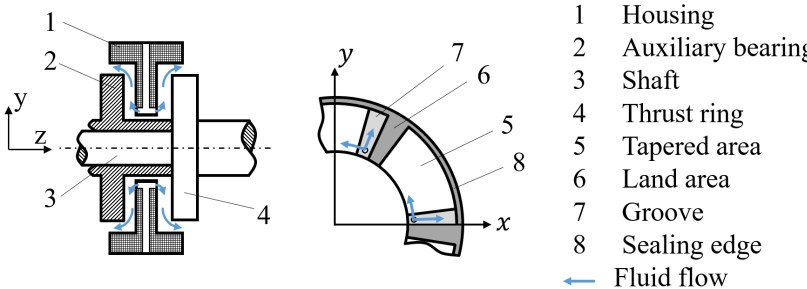

**Figure 3.** Schematic representation of the thrust bearing with considered fluid flows.

The following sections present the theoretical fundamentals for determining the pressure distribution in hydrodynamic bearings by solving the Reynolds equation taking mass-conserving cavitation into account. Besides the influence of cavitation, thermodynamic processes in the lubricating film and at supported elements as well as their coupling to each other are explained more in detail.

### 2.1. Time Integration

With respect to the simulation of transient processes, time integration and the implementation of hydrodynamics is described first. Within the multi-body system (MBS), the differential equation of motion

$$\underline{\underline{M}}(\underline{y}) \cdot \underline{\ddot{y}} + \underline{h}_\omega(\underline{\dot{y}}) + \underline{h}_{el}(\underline{y}, \underline{\dot{y}}) = \underline{h}_e(t, \underline{y}, \underline{\dot{y}}) \tag{1}$$

is solved, where $\underline{\underline{M}}$ is equal to the inertia matrix and $\underline{y}$ represents the state vector of the bushings and the degrees of freedom of the elastic rotor. In addition, the gyroscopic effects, centrifugal and Coriolis forces $\underline{h}_\omega$ as well as the stiffness and damping properties of the shaft $\underline{h}_{el}$ are taken into account. External forces such as the unbalance or resulting forces of the hydrodynamics are summarised in $\underline{h}_e$. The workflow of time integration is shown in Figure 4.

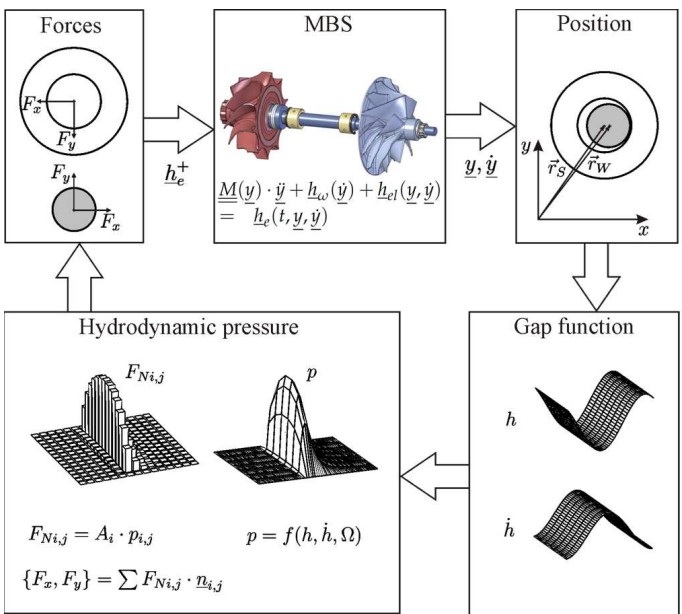

**Figure 4.** Workflow of time integration, according to Woschke [1].

Starting from the rotor position at the current time, the lubrication gap $h$ und its time derivation $\dot{h}$ can be determined via the kinematic states of the shaft and housing ($\vec{r}_S, \vec{r}_H$). With knowledge of the gap, the hydrodynamic pressure distribution is calculated by solving the Reynolds equation. Subsequently, an integration of the pressure over the bearing surface provides the resulting bearing forces acting on the shaft and housing ($F_x, F_y$). With regard to the thrust bearing, both thrust forces and torques are considered due to the asymmetrical pressure distribution. With knowledge of the forces and torques acting on the rotor, the right-hand side of Equation (1) and finally the acceleration vector can be calculated. By executing a time integration procedure, the state vector $(\underline{y}, \underline{\dot{y}})$ at the next time step is known.

The described procedure of time integration corresponds to an online solution of the Reynolds equation. The pressure distribution is calculated at each time step, which provides the advantage of an exact determination of bearing forces depending on the current kinematic state of the supporting elements. Consequently, it is not necessary to generate stiffness and damping tables and interpolate the table data to calculate the acting forces.

### 2.2. Reynolds Equation for Journal and Thrust Bearings

To determine the pressure distribution in journal bearings, a simplified representation of the Navier–Stokes equation is used [1,2,26]. It is assumed that the lubricant is incompressible and has a laminar flow behaviour. Furthermore, the oil properties can be variable due to the lubricant film temperature. The implementation of the bearing-specific assumptions leads to the generalised Reynolds equation [12,24] for journal bearings

$$0 = \underbrace{\frac{\partial}{\partial x}\left(G\frac{\partial p}{\partial x}\right) + \frac{\partial}{\partial y}\left(G\frac{\partial p}{\partial y}\right)}_{\text{Poiseuille-flow}} + \underbrace{\frac{\partial}{\partial x}\left(F_{CS}u_S + F_{CH}u_H\right)}_{\text{Couette-flow}} + \underbrace{\frac{\partial}{\partial t}\left(F_\rho\right)}_{\text{Squeeze-film flow}} \quad (2)$$

with

$$F_0 = \int_{z=0}^{h}\frac{1}{\eta}dz\,, \quad F_1 = \int_{z=0}^{h}\frac{z}{\eta}dz\,, \quad I_0 = \int_{z=0}^{z}\frac{1}{\eta}dz\,, \quad I_1 = \int_{z=0}^{z}\frac{z}{\eta}dz \quad (3)$$

and

$$G = \int_{z=0}^{h} \rho \left[ I_1 - \frac{F_1}{F_0} I_0 \right] dz \,, \tag{4}$$

$$F_{CS} = \int_{z=0}^{h} \rho \frac{I_0}{F_0} dz \,, \tag{5}$$

$$F_{CH} = \int_{z=0}^{h} \rho \left( 1 - \frac{I_0}{F_0} \right) dz \,, \tag{6}$$

$$F_{\rho} = \int_{z=0}^{h} \rho \, dz \,. \tag{7}$$

In Equations (2)–(7), $p$ is the hydrodynamic pressure, $h$ is the lubricant gap height and $\rho$, $\eta$ are the density and viscosity of the lubricant. Due to the assumption of a significantly smaller lubrication gap height compared to other bearing dimensions, cartesian coordinates with $x$ and $y$ can be used for the bearing circumference and width direction. Furthermore, the circumferential velocity on the surface of the shaft $u_S$ and housing $u_H$ can be used as boundary conditions for the fluid flow. The time derivative is needed to take transient effects into account.

A simplification of Equation (2) is achieved by assuming constant lubricant properties over the gap height. As a consequence, Equations (3)–(7) can be calculated, leading to the Reynolds equation

$$0 = \underbrace{- \frac{\partial}{\partial x} \left( \frac{\rho h^3}{12\eta} \frac{\partial p}{\partial x} \right) - \frac{\partial}{\partial y} \left( \frac{\rho h^3}{12\eta} \frac{\partial p}{\partial y} \right)}_{\text{Poiseuille-flow}} + \underbrace{\frac{\partial}{\partial x} \left( \rho h \frac{u_S + u_H}{2} \right)}_{\text{Couette-flow}} + \underbrace{\frac{\partial}{\partial t} (\rho h)}_{\text{Squeeze-film flow}} \,. \tag{8}$$

At this point, it should be mentioned that the lubricant properties need to be averaged over the gap height to use Equation (8). Using a film-averaged temperature to determine viscosity would lead to a reduced pressure distribution and softer bearing behaviour. For the numerical implementation, the three-dimensionally distributed lubricant properties are determined first as a function of oil temperature and subsequently averaged over the gap height. The oil properties are averaged after the non-linear properties have been taken into account.

Within this contribution, Equation (8) is chosen to determine the pressure distribution in journal bearings because the computing times are lower compared to the generalised Reynolds equation. This is due to the integrals within the generalised Reynolds equation, which have to be calculated for each time step of the time integration. To show the validity of the simplified Reynolds equation, the spectrograms for the run-up simulations with the generalised Reynolds equation are in the Appendix A.

To determine the pressure distribution in thrust bearings, the generalised Reynolds equation with centrifugal force can also be used, but, for the same reasons as for the journal bearing, the lubricant properties can be averaged over the gap height, resulting in the Reynolds equation

$$0 = \underbrace{- \frac{\partial}{r\partial r} \left( r \frac{\rho h^3}{12\eta} \frac{\partial p}{\partial r} \right) - \frac{\partial}{r\partial \varphi} \left( \frac{\rho h^3}{12\eta} \frac{\partial p}{r\partial \varphi} \right)}_{\text{Poiseuille-flow}} + \underbrace{\frac{\partial}{r\partial \varphi} \left( \rho h \frac{u_{\varphi S} + u_{\varphi H}}{2} \right)}_{\text{Couette-flow}} + \underbrace{\frac{\partial}{\partial t} (\rho h)}_{\text{Squeeze-film flow}} \tag{9}$$

$$+ \underbrace{\frac{\partial}{r\partial r} \left( \frac{\rho^2 h^3}{\eta} \left( \frac{u_{\varphi S}^2}{40} + \frac{u_{\varphi S} u_{\varphi H}}{30} + \frac{u_{\varphi H}^2}{40} \right) \right)}_{\text{Centrifugal flow}} \,.$$

Due to the bearing design, centrifugal flows have to be taken into account because they provide an additional flow component in radial direction. In general, centrifugal

flows can be induced via the hydrodynamic pressure gradient and via the circumferential velocities of the fluid. In this contribution, it is assumed that the latter influence has a more significant effect. The lubrication gap is in the fifth and seventh power in the case of the pressure-induced centrifugal flow, while the lubrication gap is in the third power when circumferential velocity is considered [38–40].

By solving the Reynolds equation, the hydrodynamic pressure distribution is known. However, cavitation phenomena occur in the area of the opening lubrication gap, which lead to the generation of a multiphase flow. With regard to the modelling of outgassing processes, the focus is on diffusion-induced gas generation with the implementation of the two-phase model. Therein, the gas phases dissolved in the lubricant and those present separately in the lubrication gap are considered. Detailed explanations of the two-phase model can be found in [23,24,41], so a summary is provided within this contribution.

### 2.3. Two-Phase Flow Cavitation

For further examinations, a control volume within a partially filled lubrication gap is considered (cf. Figure 5). The phase transition takes place at the surface of the bubble.

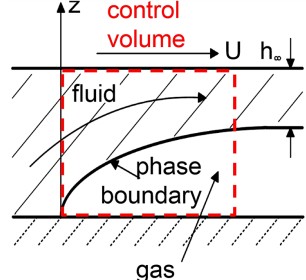

**Figure 5.** Partially filled lubrication gap with control volume, according to Dowson [20].

Based on Figure 5, it becomes clear that the lubricant properties are significant for the pressure build-up, while the properties of the gas phase can be neglected. Consequently, within the Reynolds equation (Equations (8) and (9)), effective lubricant properties

$$\eta_{eff} = \eta_{liq}F + \eta_g(1 - F) \approx \eta_{liq}F \tag{10}$$
$$\rho_{eff} = \rho_{liq}F + \rho_g(1 - F) \approx \rho_{liq}F$$

are used. The lubricant fraction $F$ represents the ratio of the oil volume (hatched area) in relation to the size of the control volume. For further calculations, the bubble content $r$ is introduced, which describes the ratio of the bubble volume to the oil volume as follows

$$F = \frac{V_{oil}}{V_{CV}} = \frac{r}{1 + r} \tag{11}$$

$$r = \frac{V_B}{V_{oil}} \quad . \tag{12}$$

The basis for implementing the two-phase model is the determination of total gas mass $m_B$, which is composed of the masses dissolved and undissolved in the lubricant

$$m_B = m_{B\,dis} + m_{B\,undis} = const \quad . \tag{13}$$

For determining the undissolved gas masses, the application of the ideal gas law (cf. Equation (14)) is sufficient, whereas, for the dissolved gas masses, the Bunsen solubility (Henry–Dalton law) has to be taken into account as well (cf. Equations (15) and (16)). The Henry–Dalton law describes the solubility of gases in lubricants and provides the possibility of idealised modelling of diffusion-induced phase transformations. The gas volume $V_{B\,dis}$ dissolved in the lubricant is linearly dependent on the existing oil volume

$V_{oil}$ and the ratio of hydrodynamic pressure $p$ to a reference pressure $p_0$. For the lubricants ISO-VG 32-220, the Bunsen coefficient is $\alpha_B = 0.08 - 0.09$ [23].

$$m_{B\,undis} = V_B \frac{p}{R\,T} \tag{14}$$

$$V_{B\,dis} = \alpha_B V_{oil} \frac{p}{p_0} \tag{15}$$

$$m_{B\,dis} = V_{B\,dis} \frac{p}{RT} = \alpha_B V_{oil} \frac{p^2}{p_0 R\,T} \tag{16}$$

Inserting Equations (14) and (16) into Equation (13) with inclusion of the bubble content by Equation (12) gives the total gas masses

$$m_B = \left( r + \alpha_B \frac{p}{p_0} \right) \frac{V_{oil}\,p}{R\,T} = const. \quad . \tag{17}$$

The assumption of a constant total gas mass provides the possibility to compare the current operating bearing condition with a reference one [24,41]. With regard to the reference condition, the pressure $p_0$, lubricant film temperature $T_0$ and reference bubble content $r_0$ are known, so the total mass of gas can be calculated via Equation (17). With knowledge of the current operating bearing state (hydrodynamic pressure $p$ and oil temperature $T$ at the current step of time integration), the current bubble content and lubricant fraction can be determined

$$m_{B\,0} = m_B \tag{18}$$

$$r = r_0 \frac{p_0 T}{p T_0} - \alpha_B \frac{p T_0 - p_0 T}{p T_0} \tag{19}$$

$$F = \frac{p}{(r_0 + \alpha_B)p_0 \dfrac{T}{T_0} + (1 - \alpha_B)p} \quad . \tag{20}$$

With the occurrence of gas cavitation, a partially filled lubrication gap is established, which depends on the hydrodynamic pressure and lubricant film temperature. Restrictions of the cavitation model result from the simplified modelling of phase transitions between dissolved and undissolved gas masses. Here, the equilibrium between the gas phases is instantaneous, which means that the inertia of the bubble growth is neglected. To take the bubble inertia into account, the solution of the Rayleigh–Plesset equation is recommended. This is a differential equation derived from the Navier–Stokes equation, which determines the size of the bubbles as a function of the pressure, surface tensions and inertial effects. In terms of an efficient implementation of outgassing processes, the evaluation of the two-phase model is sufficient for the investigation of hydrodynamically supported rotors. Furthermore, it should be mentioned that surface damage caused by cavitation is not considered. The focus is on the generation of a two-phase flow.

For the numerical implementation of the two-phase model, the introduction of a pressure-related lubricant fraction

$$F_D = \frac{F}{p} = \frac{1}{(r_0 + \alpha_B)p_0 \dfrac{T}{T_0} + (1 - \alpha_B)p} \tag{21}$$

is recommended. This procedure provides the opportunity to apply stabilisation methods in the cavitation domain (First Order Upwind) and methods for efficient solution of the Reynolds equation with mass-conserving cavitation (Newton–Raphson method).

In summary, the Reynolds equation with mass-conserving cavitation according to the two-phase model for the journal bearing is

$$0 = -\frac{\partial}{\partial x}\left(\frac{\rho_{liq}\,h^3}{12\,\eta_{liq}}\frac{\partial p}{\partial x}\right) - \frac{\partial}{\partial y}\left(\frac{\rho_{liq}\,h^3}{12\,\eta_{liq}}\frac{\partial p}{\partial y}\right) + \frac{\partial}{\partial x}\left(\rho_{liq}h\frac{u_S + u_H}{2}F_D\,p\right) + \frac{\partial}{\partial t}\left(\rho_{liq}h\,F_D\,p\right) \tag{22}$$

and for the thrust bearing taking centrifugal flow into account

$$0 = -\frac{\partial}{r\partial r}\left(r\frac{\rho_{liq}h^3}{12\eta_{liq}}\frac{\partial p}{\partial r}\right) - \frac{\partial}{r\partial\varphi}\left(\frac{\rho_{liq}h^3}{12\eta_{liq}}\frac{\partial p}{r\partial\varphi}\right) + \frac{\partial}{r\partial\varphi}\left(\rho_{liq}h\frac{u_{\varphi S} + u_{\varphi H}}{2}F_D\,p\right) + \frac{\partial}{\partial t}\left(\rho_{liq}h F_D\,p\right)$$
$$+\frac{\partial}{r\partial r}\left(\frac{\rho_{liq}^2 h^3}{\eta_{liq}}\left(\frac{u_{\varphi S}^2}{40} + \frac{u_{\varphi S}\,u_{\varphi H}}{30} + \frac{u_{\varphi H}^2}{40}\right)F_D\,p\right) \quad . \tag{23}$$

### 2.4. Temperature Model

In addition to the Reynolds equation, the energy and heat conduction equations are also needed to describe the bearing condition with regard to the temperature distribution in the lubricant film and on the shaft and housing, respectively. The oil temperature is calculated by

$$\underbrace{\frac{\partial T}{\partial t}}_{\text{instat. t.}} + \underbrace{u\frac{\partial T}{\partial x} + v\frac{\partial T}{\partial y} + w\frac{\partial T}{\partial z}}_{\text{heat convection}} = \underbrace{\frac{\lambda}{\rho\,c_p}\left[\frac{\partial}{\partial x}\left(\frac{\partial T}{\partial x}\right) + \frac{\partial}{\partial y}\left(\frac{\partial T}{\partial y}\right) + \frac{\partial}{\partial z}\left(\frac{\partial T}{\partial z}\right)\right]}_{\text{heat conduction}} + \underbrace{\frac{\eta}{\rho\,c_p}\left[\left(\frac{\partial u}{\partial z}\right)^2 + \left(\frac{\partial v}{\partial z}\right)^2\right]}_{\text{lubricant film dissipation}}, \tag{24}$$

wherein $T$ is the oil temperature; $(u, v, w)$ denotes the fluid velocity in bearing circumferential, width and height direction; and $\eta, \rho, \lambda, c_p$ are lubricant properties in terms of viscosity, density, thermal conductivity and specific heat capacity. The dissipative part of Equation (24) is due to the lubricant shearing resulting from the change of the fluid flow over the gap height ($\frac{\partial u}{\partial z}, \frac{\partial v}{\partial z}$). The gradients of the velocities are already known from the derivation of the Reynolds equation

$$\frac{\partial u}{\partial z} = \frac{1}{\eta}\left[\frac{\partial p}{\partial x}\left(z - \frac{F_1}{F_0}\right) + \frac{u_S - u_H}{F_0}\right] \quad , \tag{25}$$

$$\frac{\partial v}{\partial z} = \frac{1}{\eta}\frac{\partial p}{\partial y}\left(z - \frac{F_1}{F_0}\right) \quad . \tag{26}$$

With regard to heat convection, the velocities are calculated by

$$u(z) = \frac{\partial p}{\partial x}\left[I_1 - \frac{F_1}{F_0}\cdot I_0\right] + \frac{I_0}{F_0}(u_S - u_H) + u_H \quad , \tag{27}$$

$$v(z) = \frac{\partial p}{\partial y}\left[I_1 - \frac{F_1}{F_0}\cdot I_0\right] . \tag{28}$$

It should be mentioned that effective lubricant properties, according to Equation (10), have to be used. Otherwise, there will be an overestimation of dissipation and heat generation [42]. Furthermore, the velocity component $w$ can be neglected since it is much smaller than the other ones and laminar flow is assumed.

To determine the three-dimensional temperature distribution at the shaft and housing, the heat conduction equation is evaluated

$$\frac{\partial T_{S,H}}{\partial t} = \frac{\lambda}{\rho\,c}\left[\frac{\partial}{r\partial\varphi}\left(\frac{\partial T_{S,H}}{r\partial\varphi}\right) + \frac{\partial}{r\partial r}\left(r\frac{\partial T_{S,H}}{\partial r}\right) + \frac{\partial}{\partial z}\left(\frac{\partial T_{S,H}}{\partial z}\right)\right] \tag{29}$$

in cylindrical coordinates, where $T_{S,H}$ is the temperature of shaft and housing, respectively; $r, \varphi, z$ are the coordinates in radial, circumferential and bearing width direction; and $\rho, c$ and $\lambda$ represent the material properties of the bearing parts.

### 2.5. Coupling of Thermo- and Hydrodynamics

Considering Equations (22), (24) and (29), it becomes clear that a coupled system has to be solved to describe the thermo-hydrodynamic processes in journal bearings. The interaction consists of, on the one hand, the thermodynamics and hydrodynamics within the lubricating film and, on the other hand, of the heat transfer processes between the fluid and the shaft, bushing or housing. The interrelationships and holistic interactions are shown in Figure 6.

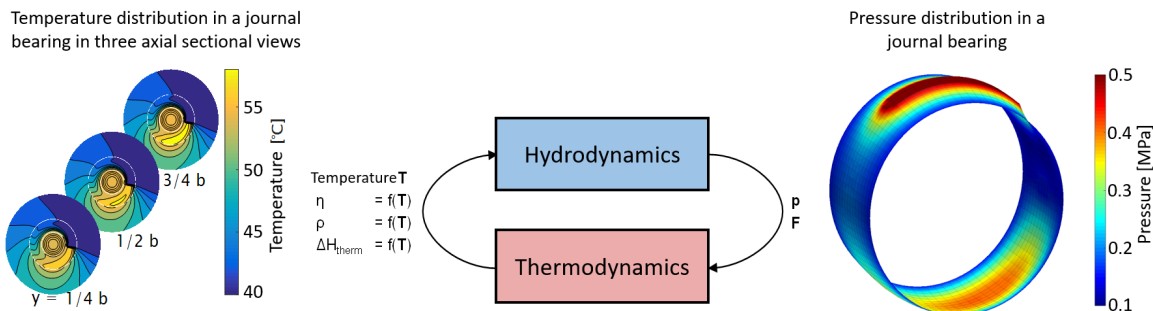

**Figure 6.** Coupling of thermodynamics and hydrodynamics.

With regard to the pressure distribution in journal bearings, Equation (22) is evaluated, wherein the hydrodynamic pressure depends on the temperature-dependent lubricant properties as well as the lubrication gap including thermal gap expansion and the oil temperature. Concerning the occurrence of outgassing processes, it should be noted that the lubricant fraction depends on both the hydrodynamic pressure and the lubricant film temperature (cf. Equation (20)). Thus, heating of the oil can lead to increasing bubble generation, so the lubricant fraction decreases and consequently outgassing processes are favoured. The occurrence of cavitation has a significant influence on the resulting forces and torques of the bearing and thus on the bearing stiffness and damping. With the solution of the Reynolds equation and two-phase model, the pressure and lubricant distributions are known so that the thermodynamic states can be determined as the next step.

With knowledge of the pressure and lubricant fraction, the fluid velocities in the lubrication gap and their change over the gap height can be calculated, so the dissipative part and the components of the heat convection of Equation (24) are determined. Subsequently, the temperature distribution in the lubricating film as well as at the shaft and housing is evaluated, in which a thermally fully coupled system is considered. Heat transfer between fluid and solid domain is based on a conjugate heat transfer model, in which heat conduction processes at the surface of the supported components are assumed. Heat transfer coefficients, which are difficult to determine, are not needed within the temperature calculations. With the solution of temperature distribution, the viscosity, density and thermal gap changes can be updated, which are again input variables for the hydrodynamics.

With regard to the shaft motion measurements, the turbocharger run-up was carried out slowly, so the assumption of a quasi-steady-state temperature distribution is valid. Furthermore, it is shown in [42] that solving the energy and heat conduction equation is not necessary at every time step, since thermal processes take place significantly more slowly than rotordynamic and hydrodynamic processes. Furthermore, solving these equations at defined output steps provides the advantage of saving computational time. The temperature is kept constant between the output steps.

### 2.6. Validation of Hydrodynamics and Thermodynamics

#### 2.6.1. Journal and Floating Ring Bearing

Validation of the hydrodynamic model taking mass-conserving cavitation into account has already been done for a rigid shaft supported by journal bearings and for a Jeffcott rotor supported by full-floating ring bearings [41]. On the one hand, the shaft motion under

cyclic load is evaluated and compared with the results of Ausas [28]. On the other hand, the run-up behaviour of a Jeffcott rotor with floating ring bearings is investigated, whereby the rotor's response under the assumption of Half-Sommerfeld solution and the two-phase model is compared with the results of Eling [19]. The influence of cavitation is shown by the occurrence of sub-synchronous oscillations over a larger speed range. In summary, the run-up simulation with mass-conserving cavitation shows better agreement with the shaft motion measurement.

The validation of the thermodynamics is carried out in [42]. There, a Jeffcott rotor with journal bearings is examined with regard to the influence of temperature assumptions on the generation of sub-synchronous oscillations. Run-up simulations are performed assuming isothermal operating conditions, a thermal lumped mass model and the complete solution of the three-dimensional energy and heat conduction equations. In summary, the rotor response based on the solution of energy and heat conduction equations shows the best agreement with the results of Eling [19]. Furthermore, it is shown that the solution of the energy equation does not have to be calculated in every time step, so the solution at defined time steps is also sufficient.

### 2.6.2. Thrust Bearing

In this section, the results of the validation for hydrodynamics at thrust bearing are shown. For this purpose, a comparison is made with the results of Hao [38]. The corresponding experimental investigations were carried out by Zhang [43]. The considered thrust bearing consists of 24 segments, where each of them has a flat and grooved area (cf. Figure 7). Furthermore, ambient pressure is assumed at bearing edges. Under these conditions, the lubricant fraction at inner and outer radius is equal to $F = 1$, which has the consequence that the oil is supplied to the lubrication gap via the bearing edges. The bearing design and the corresponding operating boundary conditions are summarised in Tables 1 and 2.

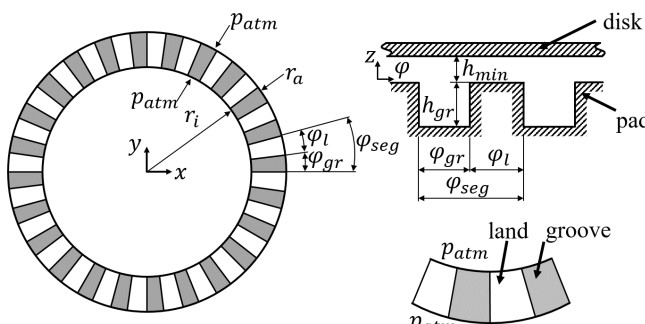

**Figure 7.** Bearing design and boundary conditions, according to Hao [38].

**Table 1.** Summary of lubrication gap, according to Hao [38].

| Property | Name | Value |
|---|---|---|
| Number of segments | | 24 |
| Inner radius | $r_i$ | 24.0 mm |
| Outer radius | $r_a$ | 32.0 mm |
| Angle of groove | $\varphi_{gr}$ | 8.4 deg |
| Angle of land | $\varphi_l$ | 6.6 deg |
| Groove depth | $h_{gr}$ | 10.2 μm |

**Table 2.** Summary of operating condition, according to Hao [38].

| Property | Name | Value |
|----------|------|-------|
| Viscosity | $\eta_{liq}$ | 0.083 Pas |
| Density | $\rho_{liq}$ | 812 kg/m$^3$ |
| Temperature | $T_{liq}$ | 22.5 °C |
| Bunsen-Solubility | $\alpha_B$ | 0.08 |
| Bubble content | $r_0$ | 0.0–0.05 |
| Angular velocity | $n$ | 333.98 rpm |
| Min. lubrication gap | $h_{min}$ | 15.4 µm |

To determine the pressure distribution, the Reynolds equation with consideration of centrifugal inertia, according to Equation (9), is evaluated for stationary operating conditions and iso-viscous temperature distributions. In [38], the implementation of mass-conserving cavitation is analogous to the principle of dissolved and undissolved gas masses explained within this contribution. However, a temperature- and pressure-dependent Bunsen coefficient was assumed, which is approximated via an exponential approach. In contrast, within this contribution the Bunsen coefficient $\alpha_B$ is constant. Furthermore, it should be mentioned that a variation of the reference bubble content is carried out in order to illustrate the influence of already existing undissolved gas masses. The results of pressure and lubricant distribution at the bearing centre plane are shown in Figure 8.

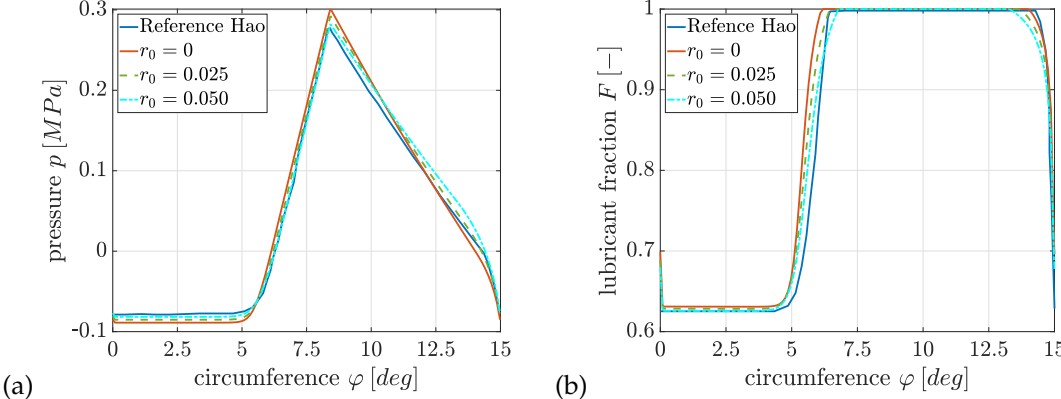

**Figure 8.** Gauge pressure (**a**) and lubricant fraction (**b**) at bearing centre plane $r = 28$ mm with $h_{min} = 15.4$ µm and $n = 333.98$ rpm.

In summary, the results show a good agreement with the determined pressure and lubricant distributions of [38]. Neglecting already undissolved air masses $r_0 = 0$, a maximum gauge pressure of $p_{max} = 0.3$ MPa is obtained between $\varphi = 6.2°$ and 13.9°. At cavitation domain, a minimum lubricant fraction of $F_{min} = 0.63$ is observed. With increasing reference bubble content, the fraction of undissolved gas masses increases, so that the maximum pressure decreases and the pressure area covers a smaller range overall. For a bubble content of $r_0 = 0.025$, the maximum pressure is $p_{max} = 0.28$ MPa. The pressure area is located between $\varphi = 6.7°$ and 13.2° and a minimum lubricant fraction of $F_{min} = 0.62$ is determined.

The pressure and lubricant distribution over the entire bearing segment is shown in Figure 9. The cavitation domain occurs mainly at groove area. Since a constant pressure is assumed at bearing edges, a pressure gradient is created at inner and outer radius so that the lubricant can be transported into the gap and thus a pressure is built up at flat area.

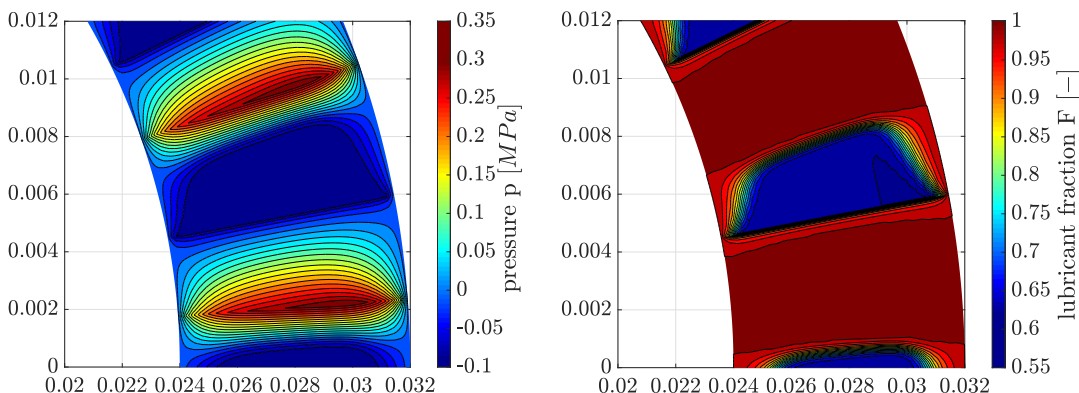

**Figure 9.** Gauge pressure (**left**) and lubricant fraction (**right**) at $h_{min}$ = 15.3 µm and $n$ = 333.98 rpm.

### 3. Results of Run up Simulations

To evaluate the rotor response behaviour, run-up simulations are compared with shaft motion measurements in the time and frequency domain. Furthermore, the influence of thrust bearings on the rotor vibrations is also examined. For this purpose, the normalised eccentricity and motion orbitals at compressor- and turbine-side bearings as well as the rotor tilting angle are discussed more in detail.

With regard to the hydrodynamic boundary conditions, constant supply pressure is assumed within the inlet holes. Due to the circumferential groove at the outer lubrication gap of the floating ring bearing, there is a constant supply pressure over the entire groove. Furthermore, the oil can flow out freely over the bearing edge since there are no lateral seals. In addition, thermal boundary conditions have to be taken into account. At the outer lubrication gap, the oil supply temperature is set within the circumferential groove. Since the angular positions of the connecting channels are the same as the positions of feeding holes and the lubricant is evenly distributed over the circumference within the groove area, an effective mixing temperature can be assumed within the supply area of the inner lubrication gap. Here, a perfect mixture of warm and fresh oil is supposed.

Besides the thermo-hydrodynamic boundary conditions, the run-up process is also considered. Within the shaft motion measurement, the run-up is relatively slow, so that the assumption of thermally steady-state operating condition is valid. In addition, thrust forces occur due to the flow of gases to the impellers. Within the simulation, the lateral forces are specified as look-up tables based on the measurement conditions. In addition, the total thrust bearing clearance is known. Within the simulation, the rotor starts from a minimum speed and accelerates constantly until the maximum speed is reached.

To evaluate the influence of cavitation and thrust bearings more precisely, simulations were carried out with various levels of detail of the bearings. As the lowest model accuracy (V1), a fully filled lubrication gap regardless of operating bearing condition (Half-Sommerfeld cavitation) is assumed and thrust bearings are neglected. An increase in the level of detail is achieved by considering outgassing processes, but thrust bearings are still neglected (V2). Consequently, a comparison of the rotor response behaviour between the model levels V1 and V2 shows the influence of cavitation. With regard to the modelling level (V3), Half-Sommerfeld cavitation is assumed again, but thrust bearings are included. As a result, a comparison between V1 and V3 shows the influence of thrust bearings on vibration behaviour. The highest model accuracy (V4) includes both mass-conserving cavitation and the consideration of thrust bearings.

#### 3.1. Shaft Motion in Frequency Domain

First, the results of the shaft motion measurement are discussed in frequency domain (cf. Figure 10). The figure shows the rotor response frequency as a function of the angular velocity. From $f_{Rot} = 0.48$ of the maximum speed, sub-synchronous oscillation with a rotor response frequency $f = 0.16$ are detected. Besides the oil-whip, the rotor behaviour is

also determined by the unbalance-induced vibrations. A synchronous resonance occurs at $f_{Rot} \approx 0.24$, whereby the amplitudes decrease after passing the resonance.

Figure 11 shows the results of the run-up simulations at various levels of detail of the bearings. Assuming a fully filled lubrication gap independent of the operating condition of the bearing and neglecting the tilting stiffness of the thrust bearings (V1), no sub-synchronous vibrations can be detected. On the one hand, a fully filled lubrication gap can lead to an overestimation of damping properties at squeeze-film damper and, on the other hand, it can lead to increased stiffness within the inner lubrication gap. Furthermore, neglecting the thrust bearing has an effect on the vibration mode of the rotor and thus also on the bearing stiffness and damping, which can be seen from the shaft orbits and the eccentricity in the bearing planes. Taking outgassing processes (V2) into account, sub-synchronous oscillation can already be predicted from a rotor speed of $f_{Rot} = 0.55$ and, thus, a better agreement with the measurements can be achieved. The occurrence of a two-phase flow leads to a reduction of bearing stiffness and damping, which favours lubricant film-induced excitations. The run-up simulations V1 and V2 were carried out without thrust bearings, with the consequence of neglecting the tilting stiffness and damping. To evaluate the influence of the tilting stiffness on the rotor response behaviour, the thrust bearing is added and Half-Sommerfeld cavitation at floating ring and thrust bearings is assumed (V3). The rotor response shows sub-synchronous oscillations from a speed of $f_{Rot} = 0.66$, whereby the oil-whip occurs later and with lower amplitude compared to the measurement. One reason for the occurrence of sub-synchronous vibrations can be that with the consideration of tilting stiffness, a rotor natural frequency is shifted closer to the half-whirl frequency of the oil at floating ring bearing. Consequently, an oil-whip can occur if a weakly-damped natural frequency of the rotor is excited. The highest level of detail is given by mass-conserving cavitation both in floating ring and thrust bearings (V4). Thus, the effects of the two-phase flow are superimposed on those of the thrust bearing, so that, in addition to the start frequency of the oil-whip $f_{Rot} = 0.50$, the rotor amplitude is also well predicted compared to the measurement. A summary of the rotor response behaviour is given in Table 3.

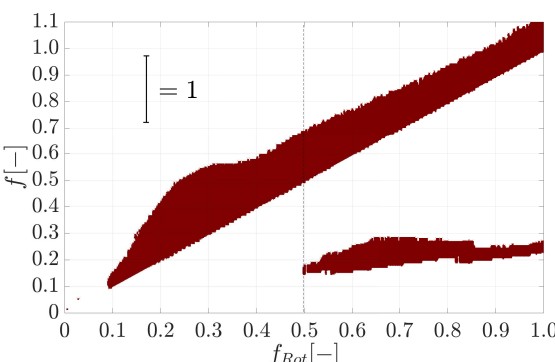

**Figure 10.** Spectrogram of the shaft motion measurement captured at the sealing disc.

**Table 3.** Summary of rotor response behaviour measured at sealing disk.

| Description | Synchronous Resonance | Start of Oil-Whip | Response Frequency |
|---|---|---|---|
| Measurement | 0.24 | 0.48 | 0.16 |
| Half-Sommerfeld without thrust bearing (V1) | 0.23 | – | – |
| 2PM without thrust bearing (V2) | 0.23 | 0.55 | 0.15 |
| Half-Sommerfeld with thrust bearing (V3) | 0.23 | 0.66 | 0.18 |
| 2PM with thrust bearing (V4) | 0.23 | 0.50 | 0.15 |

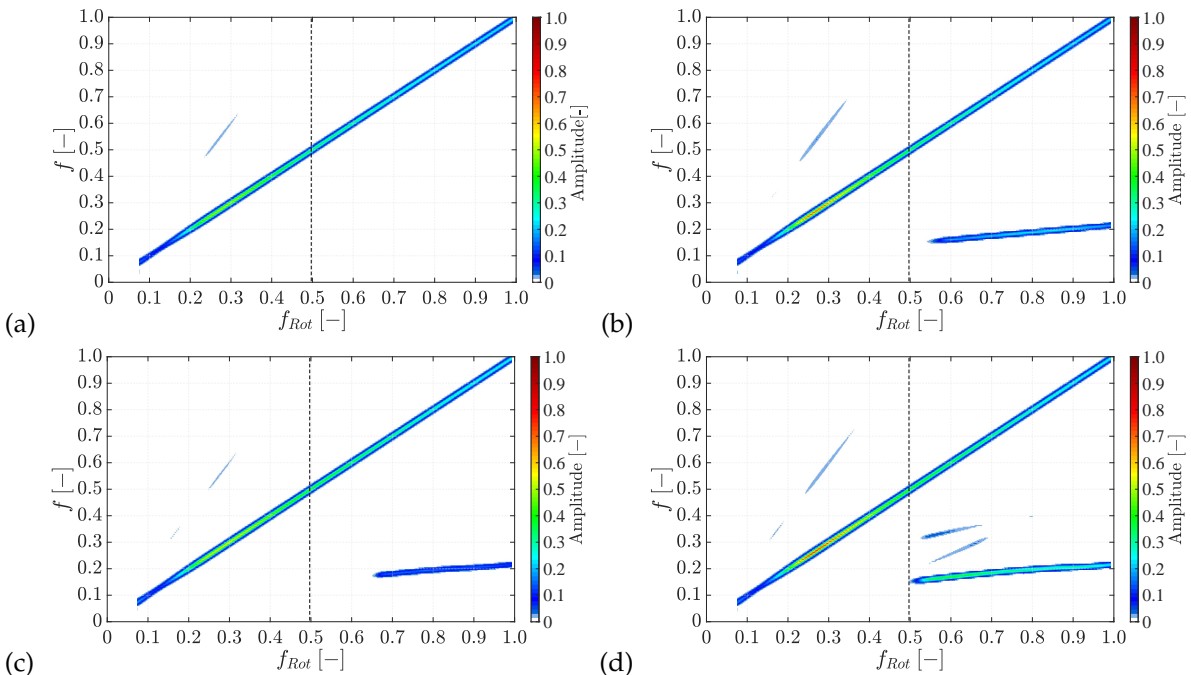

**Figure 11.** Spectrogram of the rotor response behaviour under the assumption of: (**a**) Half-Sommerfeld cavitation without thrust bearing (V1); (**b**) two-phase model without thrust bearing (V2); (**c**) Half-Sommerfeld cavitation with thrust bearing (V3); and (**d**) two-phase model with thrust bearing (V4).

### 3.2. Shaft Motion in Time Domain

Furthermore, the time signal of the horizontal rotor movement is evaluated (cf. Figure 12). The signal represents the oscillation of the rotor around its stationary position and is calculated by taking the half of the difference between upper and lower envelope curves of the time signal. Measured sub-synchronous oscillations can be observed from a speed of $f_{Rot} > 0.48$, whereby these are characterised by a stronger oscillation of the time signal. A comparison of the run-up simulations shows that a better agreement with the measurement can be achieved with increasing level of detail of the bearings. The run-up simulation with mass-conserving cavitation and without thrust bearing (V2) already showed a good congruence with respect to the start frequency of the oil-whip in the frequency domain, but the rotor oscillations are predicted too low. Finally, the simulation with consideration of thrust bearing and outgassing processes (V4) shows the best agreement with the measurement both for the start frequency of the oil-whip and with regard to the rotor oscillation magnitudes. Here, the oil-whip shows a clear increase of the vibration amplitude from a speed of $f_{Rot} > 0.50$.

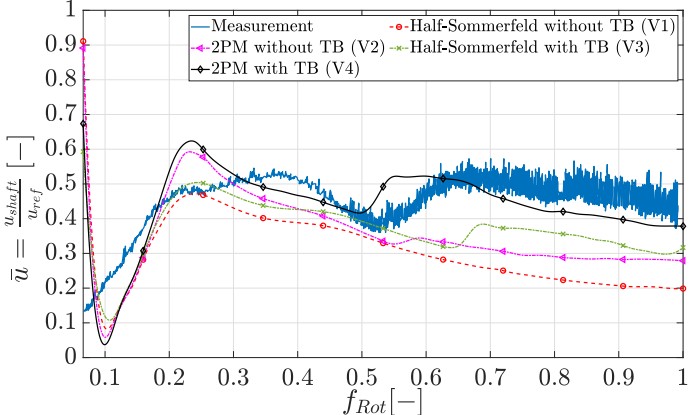

**Figure 12.** Horizontal shaft motion evaluated at sealing disc.

Discrepancies between measurement and simulations can be explained by the fact that a complete mixture between warm and cold oil is assumed within the supply area of the inner lubrication gap. A more detailed implementation of the mixing processes is based on a boundary layer model, in which a part of the warm oil can be transported over the supply area. Furthermore, it should be noted that the temperatures at thrust bearing have been specified via a look-up table. Spatial distributions of the temperatures are not taken into account. For this purpose, the three-dimensional energy equation for the lubricating film and the heat conduction equation for the shaft and housing must be solved in analogy to the thermodynamics of floating ring bearings.

### 3.3. Normalised Eccentricity and Orbits of Shaft and Floating Ring

To investigate the influence of the thrust bearing on the rotor response, the normalised eccentricities and the motion orbit of the shaft and bushings are examined (see Figures 13 and 14). For a better evaluation, simulations with mass-conserving cavitation and without thrust bearing (V2) and with thrust bearing (V4) are chosen.

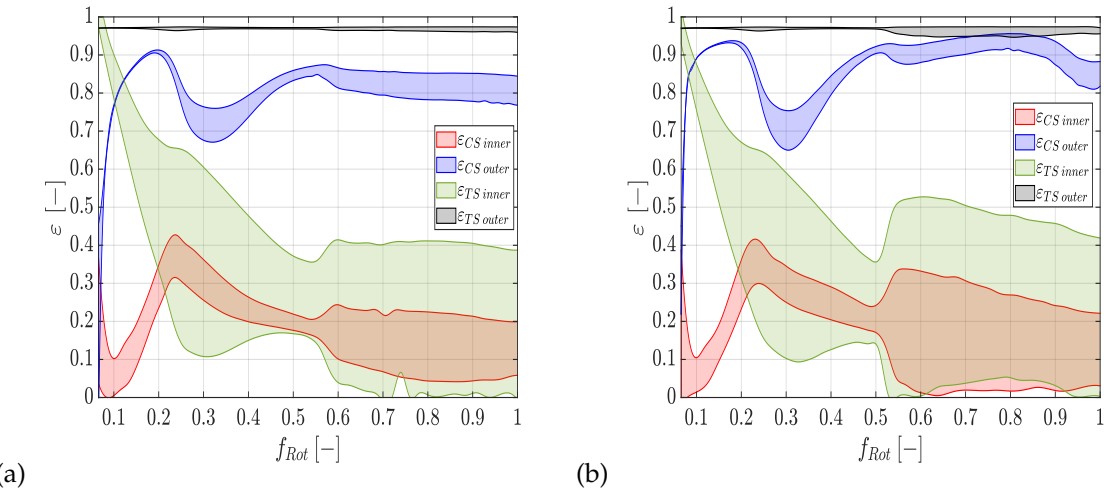

**Figure 13.** Normalised eccentricity of the shaft and bushing considering mass-conserving cavitation: without thrust bearing (V2) (**a**); and with thrust bearing (V4) (**b**). Abbreviations: *CS*, compressor side gap; *TS*, turbine side gap.

Based on the normalised eccentricity of the bushing at turbine bearing (black area in Figure 13), it can be seen that there is a contact at outer lubrication gap over the entire run-up. This is due to the rotor's design since its centre of gravity is close to the turbine bearing (see Figure 1). In addition, the outer gap is designed as squeeze-film damper since the rotational movement of the bushing is prevented. The hydrodynamic pressure can only be built up if there is a sufficient squeezing of the lubricant. Otherwise, the bushing rests. From the motion orbit of the turbine-sided bushing (grey curve in Figure 14c,d), it is also clear to see that the bushing gets immediately into contact and increased oscillation can only be observed with the occurrence of sub-synchronous oscillations at $f_{Rot} > 0.5$ or during unbalance resonance.

The influence of thrust bearings can be seen, on the one hand, with regard to the normalised eccentricity of the compressor-sided bushing (blue area in Figure 13) and, on the other hand, on the motion orbit (grey curve in Figure 14a,b). Without thrust bearings, the normalised eccentricities of the bushing can be found between $\varepsilon \approx 0.78 - 0.86$ and the contact process occurs in the upper half $\phi = 60 - 90°$. Consequently, the rotor exhibits a high degree of tilting. Considering the tilting stiffness, the rotor tilting is limited, which results in, on the one hand, higher eccentricities between $\varepsilon \approx 0.88$ and 0.95 and, on the other hand, the motion orbit of the bushing occurs at the lower region $\phi = 240–270°$. Thus, the influence of the thrust bearing is clearly illustrated by the motion orbit of the compressor-side bushing. At this point, it should be mentioned that, with increasing eccentricity of the

bushing, the anisotropy of the squeeze-film damper becomes more significant, which also has an influence on the rotor natural frequencies [10].

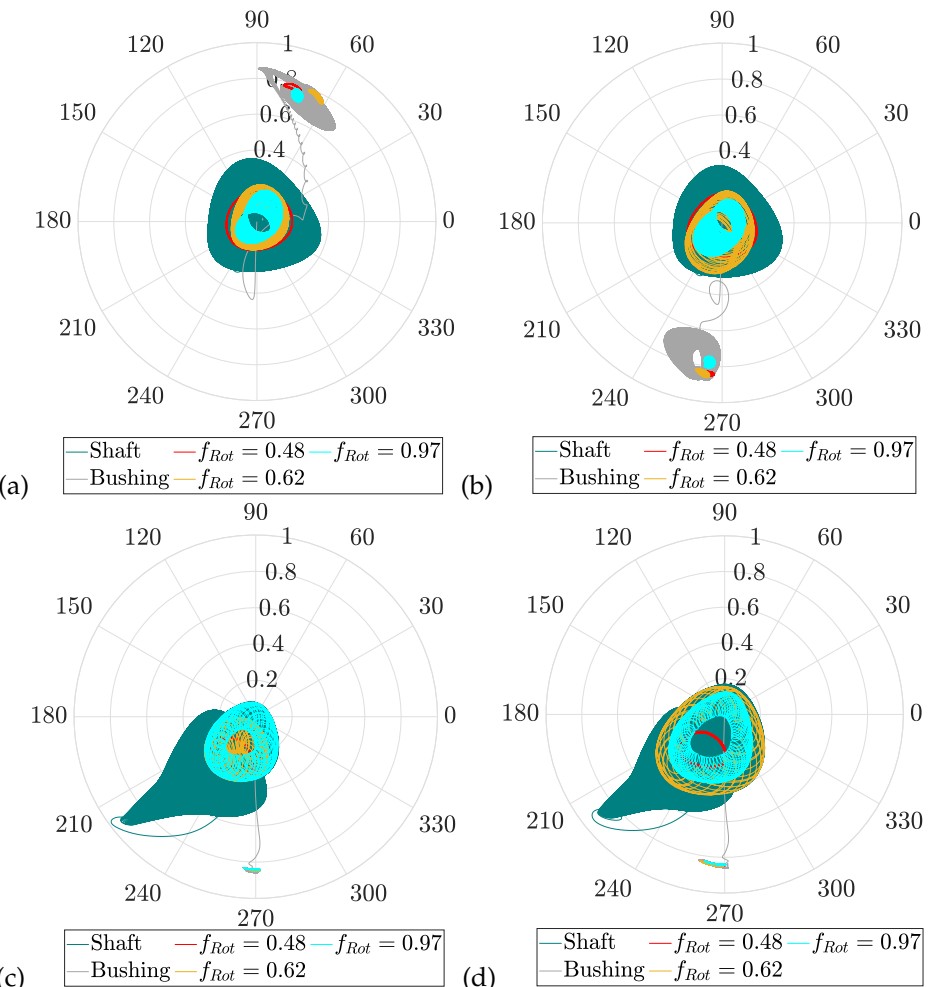

**Figure 14.** Normalised shaft and bushing orbit considering mass-conserving cavitation: (**a**) without thrust bearing at compressor bearing (V2); (**b**) with thrust bearing at compressor bearing (V4); (**c**) without thrust bearing at turbine bearing (V2); and (**d**) with thrust bearing at turbine bearing (V4).

### 3.4. Thrust Bearing

In this section, the forces and torques acting at thrust bearing are discussed (see Figures 15 and 16). With regard to the bearing forces, both thrust bearings are in equilibrium at lower speed range $f_{Rot} < 0.1$, since no thrust load is acting yet. Due to the transport of the lubricant into the narrowing gap, a hydrodynamic pressure can be built up at thrust bearing, which depends on the initial bearing clearance and thus the lubrication gap as well as on the angular velocity of the rotor. The resulting thrust forces are only equal if the thrust bearing is unloaded. Furthermore, the pre-loading of the bearing is due to the fact that the total axial bearing clearance remains constant. The axial equilibrium position of the shaft is within the bearing clearance. The axial force increases with rising rotor speed, which results in a reduction of the load on the compressor-sided and an additional load on the turbine-sided thrust bearing. The lubricant film-induced rotor excitations can also be observed in increasing oscillations of the minimum lubrication gap from $f_{Rot} > 0.52$.

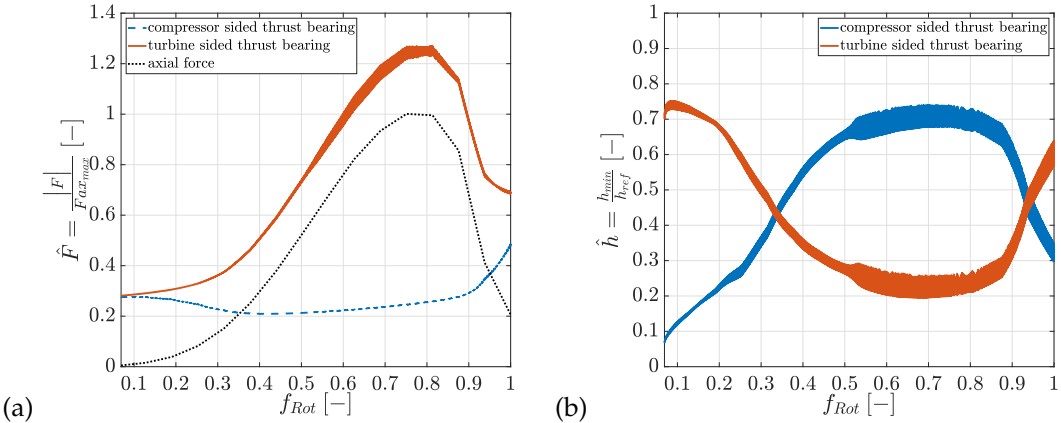

**Figure 15.** Thrust forces (**a**) and minimum lubrication gap (**b**) at thrust bearing (Bearing model level V4).

As mentioned above when evaluating the normalised eccentricity, the thrust bearing limits the rotor's tilting. For this reason, the bearing torque and the corresponding rotor tilting angle are evaluated (see Figure 16). If the thrust bearing is neglected, the rotor can tilt up to $\hat{\varphi} \approx 0.58$–$0.70$, whereas with consideration of the tilting stiffness, the rotor tilts by $\hat{\varphi} \approx 0.0$–$0.2$. The tilting angle refers to the maximum possible tilting angle of the rotor. Furthermore, the bearing torque shows that with the occurrence of sub-synchronous oscillations, increased tilting torques can occur, which contribute to the overall stiffness of the bearing.

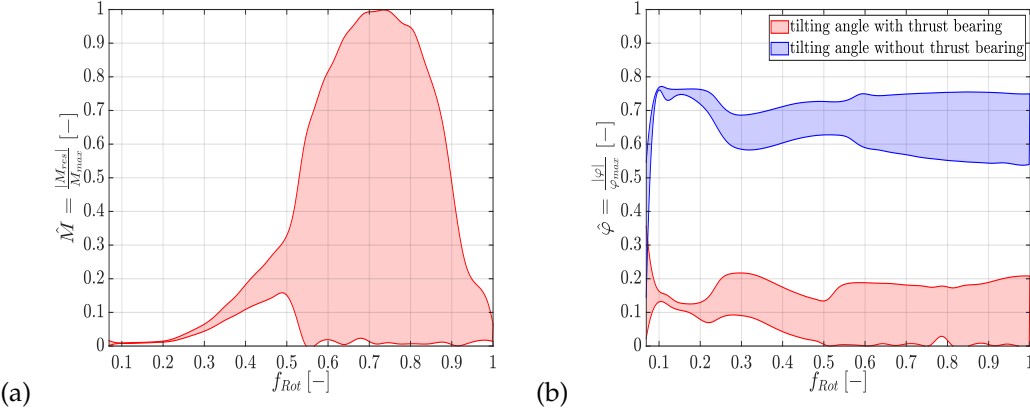

**Figure 16.** Thrust bearing torque (**a**) and tilting angle of the rotor (**b**) considering mass-conserving cavitation (V2 and V4).

## 4. Conclusions

In this contribution, the response behaviour of a semi-floating ring supported turbocharger rotor is investigated under the influence of both cavitation and thrust bearing support concerning the occurrence and amplitudes of sub-synchronous vibrations. The Reynolds equation with mass-conserving cavitation according to the two-phase model is used to determine the pressure and lubricant distribution in floating ring and thrust bearings. In addition to the hydrodynamic state, the thermodynamics at floating ring bearings are also considered. Here, the three-dimensional energy equation for the lubricant film and the heat conduction equation for the supported elements are evaluated as a fully coupled system.

To investigate the effects of cavitation and thrust bearing, various levels of detail of the bearing system are chosen and their impact on the rotor response is evaluated in time and frequency domain. As the simplest modelling, the thrust bearing is neglected and a fully filled lubrication gap (Half-Sommerfeld cavitation) is assumed (V1). Under these circumstances, no sub-synchronous vibrations could be predicted since an always fully filled gap overestimates the stiffness and damping of the floating ring bearings. A

significantly better agreement with the measurements can be achieved by taking outgassing processes into account (V2). Compared to V1, the occurrence of cavitation leads to a softer bearing behaviour with lower damping and stiffness, which favours the occurrence of oil-whip phenomena. The results with mass-conserving cavitation show better agreement with regard to the starting frequency of the oil-whip, but the vibration amplitudes are still too low compared to the measurements. Subsequently, the influence of the thrust bearing is discussed more in detail. For this purpose, run-up simulations with thrust bearing and Half-Sommerfeld solution (V3) were carried out. Here, sub-synchronous vibrations are also observed at higher speed range. The occurrence of oil-whip can be explained by the fact that the additional tilting stiffness can shift the rotor natural frequencies closer to the oil whirling frequency of the floating ring bearing. The best agreement with the measurements is achieved when both thrust bearing and outgassing processes are taken into account. Here, the effects of a softer bearing behaviour and shifting of the rotor natural frequencies are superimposed.

Furthermore, the rotor's vibration mode was investigated using the normalised eccentricity and motion orbit of the bushing and shaft. For the considered rotor, the influence of thrust bearing can be seen by the compressor sided bushing motion. If the tilting stiffness and damping are neglected, the bushing is located at the upper range of the bearing housing so that the rotor has a significant tilting position. The consideration of tilting stiffness counteracts the rotor tilting so that the motion orbit of the bushing is located in lower region. In contrast, the bushing at turbine bearing does not show any significant influence from the thrust bearing since the centre of gravity of the rotor is close to the turbine bearing and thus higher radial loads are generally present. Consequently, the influence of the thrust bearing is more significant the more decentred the centre of gravity of the rotor is, and thus a possible tilting of the rotor during turbocharger operation is favoured.

In addition to the floating ring bearing state, the minimum lubrication gap, the tilting angle of the rotor and the resulting forces and torques at thrust bearing are evaluated. The already mentioned conclusions based on the bushing movements can also be obtained when the rotor tilting angle is evaluated.

**Author Contributions:** Conceptualization, C.Z., E.W.; methodology, C.Z.; software, C.Z. and C.I.; validation, C.Z. and C.I.; formal analysis, C.Z.; investigation, C.Z. and C.I.; resources, S.N., C.D. and E.W.; data curation, C.Z.; writing—original draft preparation, C.Z.; writing—review and editing, C.Z., C.I., S.N., C.D., E.W.; visualization, C.Z.; supervision, S.N., C.D. and E.W.; project administration, S.N., C.D. and E.W.; funding acquisition, E.W. All authors have read and agreed to the published version of the manuscript.

**Funding:** The research project (FVV project no. 1258) was performed by the Junior Professorship Fluid Structure Interaction in Multibody Systems (FSK) at the Institute of Mechanics of the Otto von Guericke University Magdeburg under the direction of Jun.-Prof. Dr.-Ing. Elmar Woschke and by the Chair of Technical Dynamics (LTD) at the Institute of Mechanics of the Otto von Guericke University Magdeburg under the direction of Prof. Dr.-Ing. habil. Jens Strackeljan. Based on a decision taken by the German Bundestag, it was supported by the Federal Ministry for Economic Affairs and Energy (BMWi) and the AIF (German Federation of Industrial Research Associations eV) within the framework of the industrial collective research (IGF) programme (IGF No. 18760 BR). The project was conducted by an expert group led by Dipl.-Ing. Thomas Klimpel (ABB Turbo Systems AG). The authors gratefully acknowledge the support received from the funding organisations, from the FVV (Research Association for Combustion Engines eV) and from all those involved in the project.

**Institutional Review Board Statement:** Not applicable.

**Informed Consent Statement:** Not applicable.

**Data Availability Statement:** Not applicable.

**Conflicts of Interest:** The authors declare no conflict of interest.

**Nomenclature**

| | |
|---|---|
| $c_p$ | heat capacity |
| $f$ | ratio of the rotor response frequency and maximum rotational frequency |
| $f_{Rot}$ | ratio of the rotor speed and maximum speed |
| $F$ | lubricant fraction |
| $F_C$ | coefficient of the Couette flow |
| $F_D$ | pressure-related lubricant fraction |
| $F_{x/y}$ | bearing forces |
| $F_\rho$ | coefficient of the squeeze-film flow |
| $F_0, F_1$ | integrals for the Reynolds equation |
| $\hat{F}$ | normalised thrust force |
| $G$ | coefficient of Poiseuille flow |
| $h$ | lubricant gap height |
| $\underline{h_e}$ | external forces |
| $\underline{h_{el}}$ | elastic properties of the shaft (FEM) |
| $\underline{h_\omega}$ | gyroscopic and inertia forces |
| $\hat{h}$ | normalised axial gap |
| $I_0, I_1$ | integrals for the Reynolds equation |
| $m$ | mass |
| $\hat{M}$ | normalised bearing torque |
| $\underline{\underline{M}}$ | inertia matrix |
| $p$ | hydrodynamic pressure |
| $r$ | bubble content |
| $r, \varphi, z$ | lubrication gap coordinates |
| $\overrightarrow{r}$ | position vector |
| $R$ | universal gas constant |
| $t$ | time |
| $T$ | lubricant temperature |
| $u_{C,T}$ | unbalance |
| $u_{ref}$ | reference shaft displacement |
| $u_{shaft}$ | horizontal shaft motion |
| $\bar{u}$ | normalised horizontal shaft motion |
| $u, v, w$ | velocity components of the lubricant flow |
| $V$ | volume |
| $\underline{y}$ | state vector |
| $x, y, z$ | lubrication gap coordinates |
| $\alpha_B$ | Bunsen coefficient |
| $\varepsilon$ | normalised eccentricity |
| $\eta$ | viscosity of the lubricant |
| $\lambda$ | thermal conductivity |
| $\rho$ | density |
| $\hat{\varphi}$ | normalised tilting angle |
| $atm$ | atmosphere/ambient |
| $B$ | bubbles |
| $C$ | compressor |
| $CV$ | control volume |
| $dis$ | dissolved |
| $eff$ | effective properties |
| $g$ | gaseous phase |

| $H$ | housing |
| *min* | minimum |
| *liq* | liquid phase |
| $S$ | shaft |
| $T$ | turbine |
| *undis* | undissolved |
| 0 | reference state |

**Appendix A**

The run-up simulations shown in Section 3.1 are based on the Reynolds equation (Equations (8) and (9)), where lubricant properties are averaged over the gap height. To consider the non-linear oil properties more accurately, the run-up simulations were repeated with generalised Reynolds equation (see Figure A1). In summary, the conclusions can also be adopted here. With increasing level of detail, better agreement with the measurements can be achieved. Consequently, the run-up simulation with two-phase model and thrust bearing shows the best agreement with the measurements. Compared to the simplified Reynolds equation, the overall simulation time was longer by a factor of 10. The rotor response behaviour is summarised in Table A1.

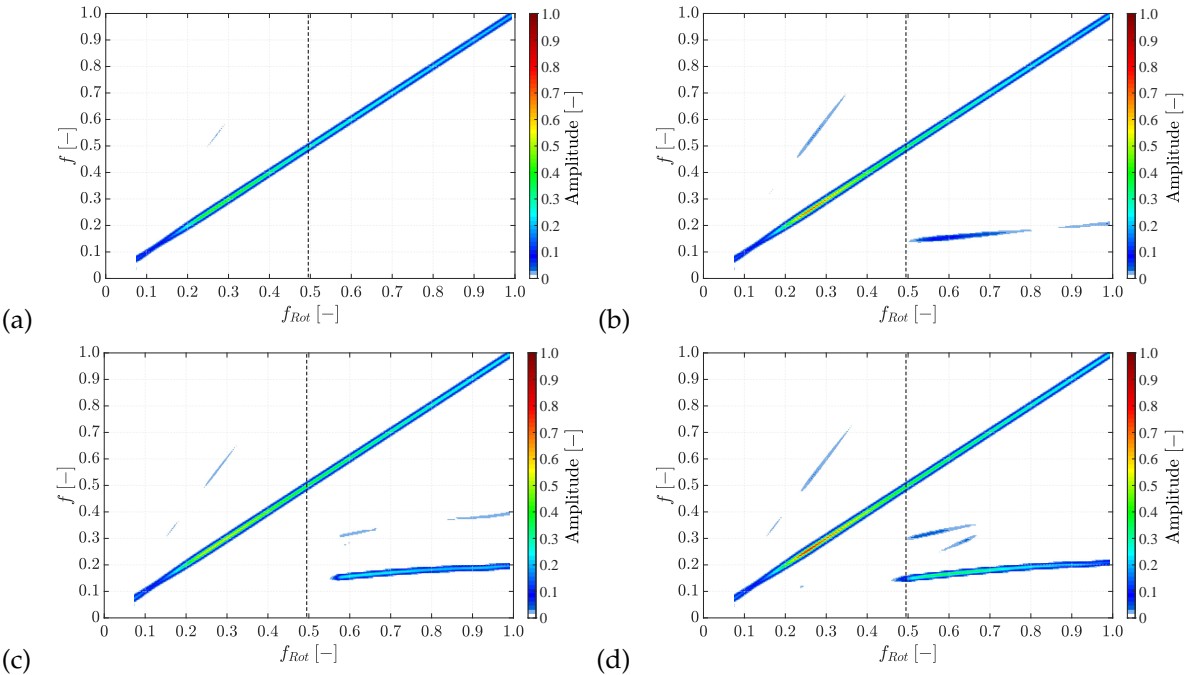

**Figure A1.** Spectrogram of the rotor response behaviour under the assumption of: (**a**) Half-Sommerfeld cavitation without thrust bearing (V1); (**b**) two-phase model without thrust bearing (V2); (**c**) Half-Sommerfeld cavitation with thrust bearing (V3); and (**d**) two-phase model with thrust bearing (V4). Run-up simulations were carried out with the generalised Reynolds equation.

**Table A1.** Summary of rotor response behaviour measured at sealing disk (Generalised Reynolds equation).

| Description | Synchronous Resonance | Start of Oil-Whip | Response Frequency |
|---|---|---|---|
| Measurement | 0.24 | 0.48 | 0.16 |
| Half-Sommerfeld without thrust bearing (V1) | 0.24 | — | — |
| 2PM without thrust bearing (V2) | 0.23 | 0.51 | 0.14 |
| Half-Sommerfeld with thrust bearing (V3) | 0.24 | 0.56 | 0.15 |
| 2PM with thrust bearing (V4) | 0.23 | 0.47 | 0.14 |

Discrepancies between the run-up simulations with simplified and generalised Reynolds equation result mainly from the integration of the non-linear oil properties via the clearance

height. The generalised Reynolds equation takes these into account more precisely, which has an effect on the pressure build-up, the resulting bearing forces and finally the response behaviour of the rotor. Nevertheless, the run-up simulations with generalised and simplified Reynolds equation show similar results. One reason for this could be that the run-up was carried out with high oil inlet temperatures (as in the measurement). Thus, the oil viscosity is in a range where temperature changes cause only minor changes in viscosity. More detailed investigations of the temperature influence on the pressure distribution are described in [17,44–46].

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
