# Peer review of "Run-Up Simulation of a Semi-Floating Ring Supported Turbocharger Rotor Considering Thrust Bearing and Mass-Conserving Cavitation"

_lubricants, doi:10.3390/lubricants9040044_

Round 1
Reviewer 1 Report
Dear Authors!
You presented very detailed results of your research aimed at solving an exclusively applied problem. The manuscript is well written, but a little overwhelmed in my opinion. I recommend to short it. Relying on the physics of the process, when solving differential equations, you are faced with the fact that the process conditions must change, so differential equations become degenerate. You don't pay your attention to this fact. It would be nice to reveal it.
Good luck
Reviewer 2 Report
The general opinion of a paper entitled „Run-up simulation of a semi-floating ring supported turbocharger rotor considering thrust bearing and mass-conserving cavitation”
The structure of the paper is correct, subsequent chapters, as well as the figures and equations, are correctly numbered. The references in the text are quoted correctly. The assessment of the grammar as well as language style I left for native speaker into consideration.
Apart from the general opinion, the reviewer has some question for the authors, which should be clarified before publications:
- At the beginning of chapter 2 “Theoretical fundamentals” the authors inform the reader, that in order to determine the elastic deformation of the shaft, the Timoshenko beam theory was used, and the problem was solved by FEM. The basic question is: how the authors simulated the supports of such beams? As a deformable or as a rigid?
- Probably by FEM analysis, the authors determined the shaft’s stiffness. How value was obtained? In formula (1) this parameter is defined as a shaft’s stiffness hel, so it should be strictly defined.
- Another component included in the formula (1) is hω. Besides of gyroscopic effects and Coriolis forces, the parameter hω should include the damping effects. This is right?
- What means the abbreviation MBS included in Figure 4?
- What means the parameters y=1/4b, 1/2b, and 3/4b included in Figure 6? Moreover, on the right side of this figure, there is an unknown element – without any description and value legend.
- What is the definition of the parameter named circumference - included in Figure 11. This parameter is ranging from 0 to 1. This is the relative value of the angle?
- In Figure 12 the authors present a spectrogram of the shaft motion determined experimentally. It would be very favorable to present and describe the test rig.
- The subsections “Shaft motion in frequency domain”, “Shaft motion in time domain”, “Normalized eccentricity and orbits of the shaft and floating ring, as well as “Thrust bearing” should be numbered as 3.1, 3.2, 3.3, and 3.4 respectively.
Reviewer 3 Report
This article has some serious problems. The whole article should be rewritten to fulfill the scope of Lubricants. The major problems are as follows:
- The abstract doesn’t reveal the objectives, method and important findings. It is too easy to be misunderstood as a review paper from this abstract.
- The introduction should be closely tied to the study background, motivation and objectives. The current version spent too much space on discussing the well-known knowledge in tribology.
- Since this article claimed that it considered the thermohydrodynamics, literature survey about the thermohydrodynamics, particularly about the modification of Reynolds equation, should be done and discussed in the introduction.
- There are too many terms that not used in tribology. For example, the radial bearing, bearing partner, outgassing, etc., just to name a few.
- The outgassing is not an appropriate term. This study is for a turbocharger under ordinary environment. There is nothing to do with the server environment that the chemical compound becomes an important issue. Outgassing is not an issue. Cavitation is.
- Word choice: Should choose the more specific terms rather than the general terms. For example, the bearing partner is a general term. It should be replaced with the name of the supported element, like shaft or journal.
- English writing: It is hard to read. For example, how a fluid flow be “concave” (line 36)? What does that mean by “to investigate the individual effects … different of levels of detail … are chosen …”?
- There is no discussion about the stiffness coefficients and damping coefficients.
- Thermohydrodynamics is a three dimensional phenomenon. The cross-film temperature variation is an important factor. The Reynolds equation needs to be re-derived and its final form should contain some integrals of fluidity (the inverse of viscosity). Using the film-averaged temperature to determine the viscosity and using this value in the ordinary Reynolds equation is not a correct approach. It will significantly underestimate the pressure reduction caused by thermohydrodynamics. The authors should use the correct version of Reynolds equation, or prove the correctness of their model.
- The Reynolds equation is a two-dimensional pressure equation. The energy equation is a three dimensional temperature equation. A three-dimensional velocity field must be determined prior to solving the energy equation. The authors should discuss how they determine the velocity components u, v and w.
Round 2
Reviewer 3 Report
There are several serious issues in this revised version:
- The authors have run the general Reynolds equation and discussed the validity of using the simplified Reynolds equation in Appendix A. This comparison however is limited as the degree of validity for their simple model depends on the viscosity-temperature coefficient. From many earlier papers discussing THD effects, they show that for high viscosity-temperature coefficient or high speed, the general Reynolds equation should be used [1]-[4]. The reviewer recommends to provide the value of viscosity-temperature coefficient and look at the Ec value [3][4].
[1] Journal of Tribology 108 (1986) 219-224
[2] Journal of Tribology 117 (1995)369-378
[3] Tribology International 41 (2008) 493–501
[4] Tribology Transactions 53 (2010) 948-956
- Clear definition should be given for the non-dimensional variables. Perhaps a nomenclature will do.
- Terminology should be tribologically specific and be consistent. For example, “radial bearing” should change to “journal bearing”, or more specific, “floating-ring journal bearing”, just to name a few.
- Fig 2 shows that the floating rings are pinned to the housing. Then how can the floating ring move and why there is an eccentricity for floating ring?
- Since the shaft is elastic, the authors should provide the material properties of shaft.
- Line 154, “In Eq. 2 and Eq. 3-7, p is equal to the …” “equal to” should be deleted.
- Line 155—158, “Due to … for the fluid flow.” The reviewer cannot understand what they mean.
- P8, sentence above Eq. (18): “With knowledge of the current …” What current?
- (21): What is D?
- Line 215, “…temperature-dependent lubricant properties…” What exactly this dependency? eta(T)=?
- Line 229: “… at the surface of the supporting components are …” What about the “supported components”?
- Line 246: “Half-Sommerfeld…” The authors should explain what the role of this model plays in here. Otherwise, the reader might misunderstand that the half-Sommerfeld is use throughout their study.
- Table 2: Should provide viscosity-temperature coefficient.
- 8: Should explain why there are differences between the curves.
- The symbol f is obvious a dimensionless parameter. Please provide its definition. Same goes to the horizontal displacement.
- 14: Except for the line colors labeled with “Shaft” and “Bushing”, the line color for the “f_Rot” on the upper row in the legend is the same as that on the lower row. Besides, what’s the difference between those two f_Rot?
- The style must be consistent in the reference list.
Author Response
Please see the attachment
Thank you very much for your comments.

Round 3
Reviewer 3 Report
All the comments have been addressed appropriately.